# Increased Red Blood Cell Distribution Predicts Severity of Chronic Obstructive Pulmonary Disease Exacerbation

**DOI:** 10.3390/jpm13050843

**Published:** 2023-05-17

**Authors:** Elias Saad, Basheer Maamoun, Assy Nimer

**Affiliations:** 1Azrieli Faculty of Medicine, Bar-Ilan University, Safad 1311502, Israel; 2Department of Medicine, Galilee Medical Center, Nahariya 2210001, Israel

**Keywords:** RDW, MPV, MCV, COPD exacerbation

## Abstract

Introduction: Increased red blood cell distribution width (RDW) has been reported to be related to underlying chronic inflammation. Our aim is to investigate the relationship of different complete blood count (CBC) parameters such as hemoglobin level, mean corpuscular volume (MCV), mean platelet volume (MPV) or RDW with COPD exacerbation severity. Methods: In the present retrospective analysis, consecutive patients admitted with the diagnosis of “COPD Exacerbation” between 1 January 2012 and 31 December 2015 were evaluated. Results: The study population included 804 patients with COPD exacerbation. The maximal partial pressure of carbon dioxide in the arterial blood (PaCO_2_) during hospital stay was significantly higher in patients with high MCV (*p* < 0.001), and in patients with a high RDW (*p* < 0.001). The hospitalization duration was significantly longer in patients with high RDW (*p* < 0.001) and in patients with elevated C-reactive protein (CRP) levels (*p* < 0.001). CRP levels strongly correlated to RDW (*p* = 0.001). Conclusions: Our study demonstrated that different CBC parameters, such as MCV and RDW, are in correlation with the severity of acute exacerbation of COPD reflected by the PaCO_2_ level and the duration of hospitalization. Furthermore, we also found a positive correlation between RDW and CRP levels. This finding supports the hypothesis that RDW is a good biomarker of acute inflammation.

## 1. Introduction

Chronic obstructive pulmonary disease (COPD) is the third leading cause of death worldwide, and it represents a major health, social and economic burden [1].

The GOLD defines an exacerbation COPD as “an event characterized by dyspnea and/or cough and sputum that worsen over ≤14 days, which may be accompanied by tachypnea and/or tachycardia and is often associated with increased local and systemic inflammation caused by airway infection, pollution, or other insult to the airways” [2]. Causes of exacerbation of COPD are multiple and include infections, heart failure, pulmonary embolism, and gastroesophageal reflux [3,4,5,6]; of these causes, infections are the most common [7]. These infective episodes may be due to bacterial, viral, and, rarely, fungal infections [8]. Quantification of the severity exacerbations in COPD is usually made clinically and is based on the symptoms, physical signs, and response to therapy. Pulmonary function tests (PFTs) have no role in assessing the severity of COPD exacerbation, although they are the cornerstone of the diagnostic evaluation of patients with suspected COPD [2,9,10], and at present, there is no method for quantifying the severity of COPD exacerbations.

Cardiovascular disease (CVD)-related morbidity and mortality is high in COPD patients [11], and more patients with COPD die from cardiovascular causes than from respiratory failure.The most obvious explanation for the high cardiovascular morbidity and mortality rates seen in COPD patients is the high prevalence of smoking and other known risk factors for CVD, such as sedentary lifestyle as well as systemic inflammation due to oxidative stress and chronic hypoxia [12].

Red blood cell distribution width (RDW) is a numerical measure of the size variability of circulating erythrocytes. RDW is in a standard size, but disorders related to systemic inflammation, ineffective erythropoiesis, nutritional deficiencies, bone marrow dysfunction or increased destruction cause a higher RDW [13,14,15,16,17]. Increased RDW values have been reported to be related to underlying chronic inflammation which promotes red blood cell membrane deformability and changes in erythropoiesis [18]. It has been defined as a prognostic tool in different clinical settings such as pulmonary arterial hypertension, congestive heart failure and coronary heart disease [19,20,21].

Increased inflammation in the lungs, as well as a systemic inflammatory response, is now a well-established factor in COPD [22,23]. A number of inflammatory markers have been shown to be elevated systemically in COPD including Interleukin (IL)-6, IL-8, Tumor necrosis factor (TNF)-α, C-reactive protein (CRP), and fibrinogen [23,24,25]. Inflammatory cytokines have been found to inhibit erythropoietin-induced erythrocyte maturation, which is reflected in part by an increase in RDW [18]. COPD-related inflammation may also impair erythropoiesis, as do other chronic inflammatory processes, and increase RDW. Although the relationship between RDW and survival is well recognized in CVD [26,27,28,29,30,31], it has not been previously demonstrated in COPD.

Our aim was to investigate the relationship of different CBC parameters such as hemoglobin level, MCV or RDW with COPD exacerbation severity.

## 2. Materials and Methods

This retrospective single-center study was performed in a tertiary medical center providing services to the population of Northern Israel. The study was approved by the institutional review board (IRB) of the Galilee medical center (GMC), Nahariya, Israel. Consecutive patients admitted to the internal medicine departments with the diagnosis of “COPD Exacerbation” between 1 January 2012 and 31 December 2015 were included.

Inclusion criteria were:Age older than 18 years.Patients have been diagnosed with COPD by GOLD criteria [2].The patient was admitted to the hospital with a clinical presentation which met the GOLD definition of acute COPD exacerbation [2].

The exclusion criteria are:Active cancer.The patient has a diagnosis of connective tissue disorder, inflammatory bowel disease, or hematological system diseases (such as malignancy, thalassemia, hemolytic anemia).The patient was treated with blood transfusion or anti-inflammatory drug (systemic steroids, immunosuppressive drugs) in the last 2 months.

## 3. Variables

The severity of COPD exacerbation was determined by the following parameters:The maximal partial pressure of carbon dioxide in the arterial blood (PaCO_2_) during hospital stay.The length of hospital stay.

The following variables were evaluated as predictors of severity of COPD exacerbartion: age, gender, Hemoglobin level, MCV, RDW, platelet count, mean platelet volume (MPV), PaCO_2_, oxygen saturation, and C-reactive protein (CRP).

## 4. Statistics

Data were evaluated using SPSS software for Windows version 23.0. Variables were given as the mean with a standard deviation value. Multivariate stepwise regression analysis was performed for individual variables, including CBC parameters to determine their effect in predecting severity of COPD exacerbation. Pearson’s correlation test was used to measure the strength of a linear association between laboratory parameters. Frequencies were compared using the chi-squared test. A *p*-value of less than 0.05 was considered to be significant.

## 5. Results

### 5.1. Demographic and Clinical Characteristics of the Patients

A total of 1011 patients met the inclusion criteria. Of these, 207 were excluded becuase of various reasons (malignancy, chronic inflammatory disease, hematologic disease). The remaining 804 patients (34% female) were included for study analyses. The mean age of the patients was 68 ± 13 years. In total, 39% of patients had anemia at presentation (Table 1).

### 5.2. The Correlation between Maximal PaCO_2_ during Hospitalization and CBC Parameters

In multiple regression analysis (Table 2A), two CBC parameters are associated with increased maximal PaCO_2_ during hospitalization, the MCV, and the RDW. Patients with MCV > 100 fL had significantly higher maximal PaCO_2_ (Figure 1A) than patients with normal or low MCV (62 mmHg vs. 56 mmHg vs. 52 mmHg, respectively, *p* < 0.001). Patients with a high RDW significantly a higher PaCO_2_ (Figure 1B) than patients with normal RDW (58 mmHg vs. 53 mmHg, *p* < 0.001).

### 5.3. The Correlation between Hospitalization Duration and CBC Parameters

In multiple regression analysis (Table 2B), two parameters are associated with increased hospitalization duration: the RDW and CRP level. Patients with a high RDW had significantly longer hospitalization duration (Figure 2A) than patients with normal RDW, and the higher RDW at admission the longer hospitalization duration (7.5 days for patients with RDW > 18 fL, 5.9 days for patients with RDW levels of 14–18 fL, and 5.1 days for patients with RDW < 14 fL, *p* < 0.001). Patients with elevated CRP level had significantly longer hospitalization duration (Figure 3) than patients with normal CRP level (6.1 days vs. 4.5 days, *p* < 0.001).

### 5.4. CRP and Its Correlation with CBC Papameters

In Pearson’s correlation analysis of CBC parameters with CRP levels (Table 3), CRP levels showed a strong positive correlation with RDW and CRP (Figure 3).

## 6. Discussion

The present study revealed that the severity of COPD exacerbation, reflected by the maximal PaCO_2_ during hospitalization and the duration of hospitalization, can be early predicted by simple blood tests taken at the emergency room (ER). Among patients who experience a COPD exacerbation, 9% to 31% require an ER visit and 14% to 35% require hospitalization [32,33,34]. Overall, 30-day mortality rates following hospitalization for COPD range from 3% to 9% [35] and 90-day mortality rates exceed 15% [36]. We have found that higher RDW values at admission predict higher levels of Paco_2_ during hospital stay with an average of 58 mmHg, as well as longer duration of hospitalization with an average of 7.5 days. Previous studies evaluated the RDW and its ability to predict prognosis in COPD. Seyhan et al. found that RDW levels were significant predictors of mortality in patients admitted with COPD exacerbation [37]. Sincer et al. found that Levels of RDW, obtained before echocardiography, predicted the presence of right ventricle failure in patients with COPD [38]. RDW value > 17.7 predicted the presence of right ventricular failure with sensitivity of 0.70 and specificity of 0.93, respectively. RDW is a quantitative measure of anisocytosis, and it is routinely reported as a component of the complete blood count [13]. Inflammatory cytokines have been found to suppress the maturation of erythrocytes, allowing juvenile erythrocytes to enter into the circulation and lead to an increase in RDW. We speculate that the underling inflammatory state in COPD patients which aggrevates in acute exacerbations is the main reason for elevated RDW in COPD patients. Our study revealed a positive correlation between RDW and CRP levels. This finding supports the hypothesis that RDW is a marker of inflammation, and this could be the explanation of the elevated RDW in multiple diseases associated with poor outcomes such as cardiovascular diseases and COPD. Lippi et al. found a graded association of RDW with high-sensitivity C-reactive protein (hsCRP) and erythrocyte sedimentation rate (ESR) in a large cohort of unselected adult outpatients [18].

In the current study, we found that elevated CRP levels (>5 mg/dL) at admission can also predict longer duration of hospitalization of patients with acute exacerbation of COPD. The role of CRP level as a predictor of prognosis in COPD has been widely evaluated. Dahl et al. found that baseline CRP level is a strong and independent predictor of future hospitalization and death in patients with COPD [39].

Our finding of macrocytosis (defined as MCV greater than 100 fL) as a predictor of CO_2_ retention in acute exacerbation of COPD have not been described before. Garcia et al. have reported that macrocytosis significantly correlates with dyspnea and FEV1 in ex-smoker COPD patients. Macrocytosis also predicted longer hospital stay with an average of 7.8 days. Macrocytosis occurs in approximately 3% of the general population [40]. Alcohol use accounts for the majority, followed by deficiencies in folate and vitamin B12 and medications. Other causes of macrocytosis include chronic liver disease, hypothyoroidism, *Helicobacter pylori* infection, hemolytic anemia, and active smoking [40,41,42]. What could be the explanation of macrocytosis and elevated PaCO_2_ in our study? In the current study, we did not have enough data about patients’ thyroid function, nor about serum levels of folate or Vitamin B12. The personal habbits of smoking or alcohol consumption also can not be an explanation, because there was no diffence between patients with macrocytosis and patients without macrocytosis with regard to the rate of active smoking and the rate of alcohol abuse. One possible explanation is that macrocytosis is caused by an undiagnosed chronic liver disease, and it is known that patients with cirrhosis have an acquired alteration of both innate and acquired immunity [43] and leading to immunodeficiency as well as systemic inflammation [44].

In conclusion our study demonstrated that elevated RDW levels as well as changes in MCV values in patients with COPD exacerbation were associated with disease severity reflected by the PaCO_2_ level and duration of hospitalization. Furthermore, we demonstrated a positive correlation between RDW and CRP levels, which supports the hypothesis of RDW as a marker of inflammation.

The strengths of this study were the large sample size (>800 patients), the findings of MCV > 100 fL and high RDW as predictors of CO_2_ retention, the positive correlation between RDW and CRP, and the statistical power of the results obtained. It proved that inexpensive and simple laboratory parameters can be considered as biomarkers of the severity of COPD exacerbation, and their use to escalate the treament of COPD exacerbation should be further investigated. There are several limitations of this study. First, in most of the patients we could not find data about the main parameters of severity of COPD, FEV1 and GOLD classification. Second, laboratory dates including RDW measurements were only available at baseline; thus, it is unknown whether RDW improves as the clinical status of the patient improves. Third, we did not evaluate if there is a correlation between CBC parameters and respiratory tract infections, which is considered the main cause of COPD exacerbations. Fourth, this study is a retrospective study, and the patients were evaluated using file records for COPD.

## Figures and Tables

**Figure 1 jpm-13-00843-f001:**
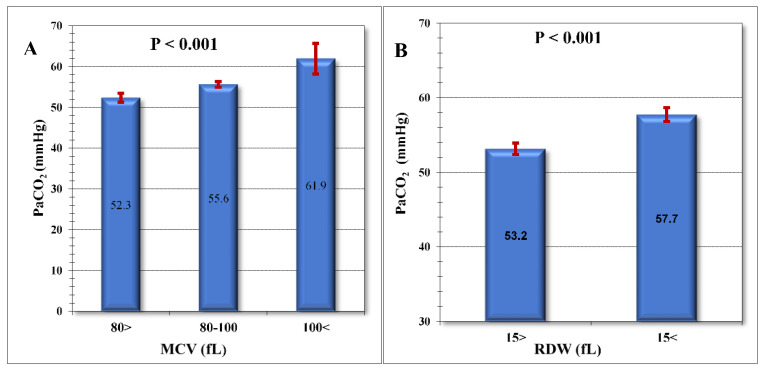
Comparison of mean levels of maximal PaCO_2_ during hospitalization and CBC parameters. (**A**) Mean levels of maximal PaCO_2_ during hospitalization in patients with macrocytosis, microcytosis or normal MCV. (**B**) Mean levels of maximal PaCO_2_ during hospitalization in patients with normal RDW or high RDW. Abbreviations: PaCO_2_, Partial Pressure of Carbon Dioxide; MCV, mean corpuscular volume; RDW, red cell distribution width; fL, femtoliter.

**Figure 2 jpm-13-00843-f002:**
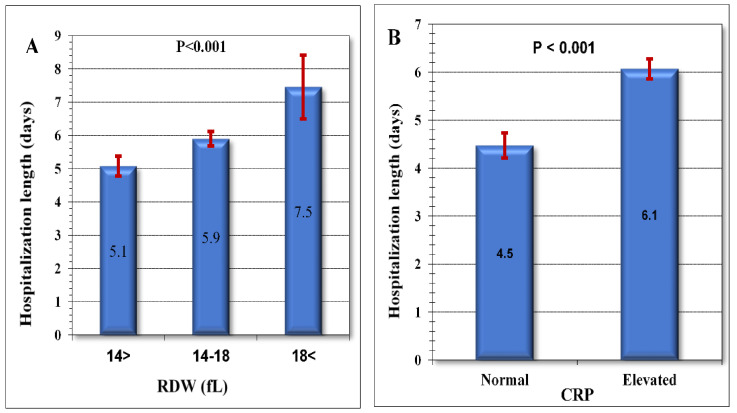
Comparison of mean hospital duration in various subgroups: (**A**) Mean duration of hospitalization (in days) in patients with Normal RDW (<14), high RDW (14–18), and very high RDW (>18). (**B**) Comparison of Mean hospitalization duration (in days) in patients with normal or elevated CRP levels (mg/dL). Abbreviations: CRP, C-Reactive Protein; RDW, red cell distribution width; fL, femtoliter.

**Figure 3 jpm-13-00843-f003:**
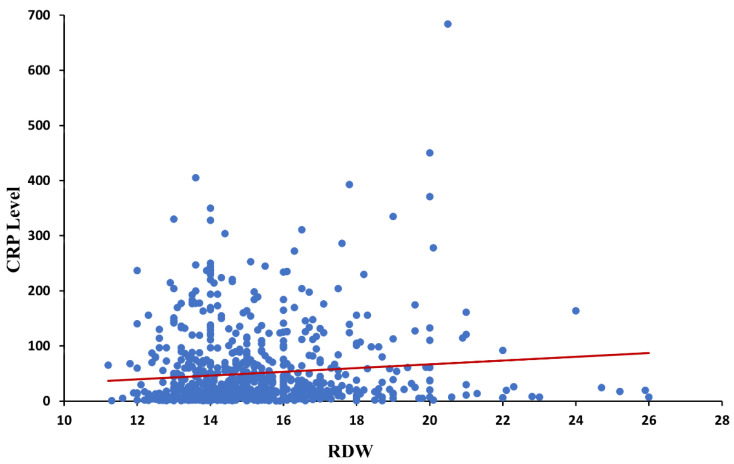
Serum CRP levels (mg/dL) plotted against RDW, in patients with COPD exacerbation. Trendline is linear. Abbreviations: CRP, C-Reactive Protein; RDW, red cell distribution width.

**Table 1 jpm-13-00843-t001:** Characteristics of study group.

	Numer of Patients
Total study group	804 patients
Men, number (%)	534 (66.4%)
Women, number (%)	270 (33.6%)
Age, mean ± SD (years)	67.7 years (±13)
Mean Age, men (year)	68.3 years
Mean Age, Women (year)	67.2 years
Patients with anemia, number (%)	317 (39.4%)
Male patient with anemia, number (%)	191 (35.8%)
Female patient with anemia, number (%)	126 (46.7%)

Abbreviations: SD, standard deviation.

**Table 2 jpm-13-00843-t002:** Multiregression analyses of the laboratory parameters their correlation to maximal PaCO_2_ during hospitalization (**A**) and to hospitalization duration (**B**).

A						
		Coefficient	95% Conf. (±)	Std. Error	T	*p* Value
	Constant					
	MCV	0.34	0.15	0.07	4.3	0.00001
	RDW	1.36	0.6	0.3	4.4	0.00001
**B**						
	Constant					
	MCV	0.35	0.15	0.07	4.6	0.00001
	RDW	0.007	0.004	0.002	3.15	0.001

Abbreviations: MCV, mean corpuscular volume; RDW, red cell distribution width; PaCO_2_, partial pressure of carbon dioxide.

**Table 3 jpm-13-00843-t003:** Pearson correlation analysis between CBC parameters and CRP level in patients with COPD exacerbation.

		Correlations				
		HB	MCV	RDW	PLT	MPV
CRP	Correlation coefficient	−0.018	0.011	0.098	0.051	0.015
	One-sided significance	0.30	0.37	0.001	0.06	0.32

Abbreviations: HB, Hemoglobin; MCV, mean corpuscular volume; RDW, red cell distribution width; MPV, mean Platelet volume; PLT, Platelet; CRP, C-Reactive Protein.

## Data Availability

The datasets used and/or analysed during the current study are available from the corresponding author on reasonable request.

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
