# Peer review of "Increased Red Blood Cell Distribution Predicts Severity of Chronic Obstructive Pulmonary Disease Exacerbation"

_jpm, 2023, doi:10.3390/jpm13050843_

Round 1

Reviewer 1 Report

Topic of great interest to potential readers of this magazine. There is no doubt that COPD is a highly prevalent disease with high morbidity and mortality and underdiagnosis. In relation to biomarkers, it is an emerging area of study and interest for all clinicians involved in the care of these patients.

However, some comments are made in favor of improving the current version of the manuscript:

.- Title. Instead of “Increased Red Blood Cell Distribution…”, perhaps better “Increased red blood cell distribution and decreased mean platelet volume predicts severity of of chronic obstructive pulmonary disease exacerbation” (thus remove COPD from the title and acronym).

.- Abstract. Consider changing “Increased Red Blood Cell Distribution” to all lower case.

Mean platelet volume” for “mean platelet volume”

maximal Partial pressure” for “maximal partial pressure”

(P<0.001)” by “(p<0.001)”

PaCO2” for “PaCO2”

are a marker of inflammation” for “are good biomarkers of acute inflammation”.

.- Introduction. Change “Background” to “Introduction”.

Review definition of COPD exacerbation. It has changed according to GOLD 2023.

.- Material and methods.

It is noteworthy that as a parameter of severity of the COPD exacerbation the duration of the average stay is taken into account. Can this variable not be influenced by multiple factors, patient age, comorbidities, healthcare pressure from the hospital, etc?

The only comorbidity mentioned in the study is anemia. Perhaps other comorbidities should have been included.

Why not take other validated severity parameters into account as clinical symptoms? Dyspnea, respiratory rate, level of consciousness….[Soler-Catalunya JJ, et al. Spanish COPD Guidelines (GesEPOC) 2021 Update Diagnosis and Treatment of COPD Exacerbation Syndrome. Arch Bronocneumol. 2022;58:159-170. DOI:10.1016/j.arbres.2021.05.033]

.- Discussion. "Literature data..." perhaps change to another more scientific expression.

Cerebrovascular ischemia” for “cebrovascular ischemia”.

.- Conclusions. Last paragraph. "The importance of this study..." is a strength in itself of this work and should be included in the Discussion section before the "limitations of the study".

Likewise, the large sample (>800 patients), the findings of MCV > 100 fL and low MPV as predictors of CO2 retention, as well as the statistical power of the results obtained, constitute other relevant strengths of this work.

.- Tables. Table 1. The standard deviation (SD) of the mean age is missing.

Other data is missing, severity of COPD according to GOLD (A,B and E), comorbidities, physical signs, etc.

Table 2. Perhaps the “correlations” field could be removed.

.- References. Extensive bibliography, which is highly appreciated [50 citations]. However, it would be convenient to review the universal literature, since none of the references is recent, that is, 5 years or less from its publication.

Appointments 1 and 27, review. They refer to GOLD 2015 and 2016. The latest update, GOLD 2023, is now available. Add access link.

It is noteworthy that 22 of the 50 citations are delimited only in the introduction section.

Author Response

dear reviewer 1

I have made changes as you suggested:

  1. first I have changed the statistical method and I used stepwise multiple regression analysis to calculate the effect of CBC parameters on COPD exacerbation parametrs and this is described in table 2.
  2. I have changed the title according to the new results
  3. I have changed the results, discussion and conclusion section according to the new analysis.
  4.  I made all language corrections you have suggested.
  5. I have changed the citations and references , I added references to many sentence in background, and I changed the references according to MDPI reference list and citations style guide 
  6. I changed the "background" to "introduction"
  7. I have updated the reference of GOLD criteria to the last updated one (2021) - reference number 2.  

Reviewer 2 Report

My comments:

                In general, this is an interesting research. The authors aim to investigate the relationship of different complete blood count (CBC) parameters such as hemoglobin level, mean corpuscular volume (MCV) or RDW with COPD exacerbation severity. However, there are major points must be improved.

Major concern

                The statistical analysis used in this study was not corrected. Due to the objective of this study, the Gaussian regression must be used for calculated the effect of  complete blood count (CBC) parameters such as hemoglobin level, mean corpuscular volume (MCV) or RDW with  COPD exacerbation severity (PaCO2 and length of stay.

Background

            - There are many sentence need references.   

Results, Discussion, and Conclusion

                The results, discussion, and conclusion sections should be changed according to new results from new analysis.

Minor

1. Title

                - Please remove “of” “Increased Red Blood Cell Distribution and decreased mean platelet  volume predicts severity of of chronic obstructive pulmonary  disease (COPD) exacerbation”

2. References

                - The references should be change according to MDPI style.

Author Response

dear reviewer 1

I have made changes as you suggested:

about major concern:  I have changed the statistical method and I used stepwise multiple regression analysis to calculate the effect of CBC parameters on COPD exacerbation parameters and this is described in table 2. and also I have added table 3 which describes the pearson correlation between CRP and CBC parameters to provide a proof that CRP correlates with RDW. I changed the results, discussion and conclusion section according to the new analysis. I changed all figures according to the new results .

about background:  I added references to many sentence in background

The tile was changed according to the new results into"Increased red blood cell distribution predicts severity of chronic obstructive pulmonary disease exacerbation"

I changed the references according to MDPI reference list and citations style guide 

Round 2

Reviewer 2 Report

Overall, the authors appropriated respond to my comments. However, there is some minor points should be improved.

1.   English language spelling check is required. For example, PCO2 or PaCO2.

2.  The strength and limitation of the study should be the last paragraph of the discussion section. 

Author Response

dear reviewer

I have made an english language spelling check and made some corrections, and I have made The strength and limitation of the study should be the last paragraph of the discussion section.